# Fatty Acid Metabolites and the Tumor Microenvironment as Potent Regulators of Cancer Stem Cell Signaling

**DOI:** 10.3390/metabo13060709

**Published:** 2023-05-31

**Authors:** Toshiyuki Murai, Satoru Matsuda

**Affiliations:** 1Graduate School of Medicine, Osaka University, 2-2 Yamada-oka, Suita 565-0871, Japan; 2Department of Food Science and Nutrition, Nara Women’s University, Kita-Uoya Nishimachi, Nara 630-8506, Japan

**Keywords:** metabolic reprogramming, lipid metabolites, tumor microenvironment, cancer stem cells, epithelial to mesenchymal transition, phosphoinositide 3-kinase, PI3K/AKT

## Abstract

Individual cancer cells are not equal but are organized into a cellular hierarchy in which only a rare few leukemia cells can self-renew in a manner reminiscent of the characteristic stem cell properties. The PI3K/AKT pathway functions in a variety of cancers and plays a critical role in the survival and proliferation of healthy cells under physiologic conditions. In addition, cancer stem cells might exhibit a variety of metabolic reprogramming phenotypes that cannot be completely attributed to the intrinsic heterogeneity of cancer. Given the heterogeneity of cancer stem cells, new strategies with single-cell resolution will become a powerful tool to eradicate the aggressive cell population harboring cancer stem cell phenotypes. Here, this article will provide an overview of the most important signaling pathways of cancer stem cells regarding their relevance to the tumor microenvironment and fatty acid metabolism, suggesting valuable strategies among cancer immunotherapies to inhibit the recurrence of tumors.

## 1. Introduction

Many research findings have suggested the importance of a small proportion of cancer cells with extraordinary resistance to the conventional cancer therapies that target the highest proportion of cancer cells [1,2,3,4]. This small cell population within a tumor, which possesses the capacity to self-renew and initiate heterogeneous lineages of cancer cells that constitute a tumor, have been defined as cancer stem cells [5]. Cancer stem cells were first identified in breast cancer; it was reported that breast tumors contained heterogeneous populations of cancer cells and that a small population with CD44-positive/high and CD24-negative/low surface expression was capable of generating tumors despite having no obvious morphologic features [6]. Thereafter, cancer stem cells were identified in other cancer types, including acute myeloid leukemia [7,8,9] and solid tumors such as breast cancer [6,10,11], pancreatic cancer [12], colon cancer [13,14,15,16], ovarian cancer [17], and a malignant brain tumor [18,19]. Hence, there is established evidence for the cancer stem cells model in which a subset of cells distinct from the bulk of those in a tumor is responsible for the long-term maintenance of tumor growth [20]. Cancer stem cells possess normal adult stem cell-like features, i.e., great potential to quickly repopulate themselves and the ability to metastasize to distant sites, forming additional lesions. Cancer stem cells were supposed to arise from normal stem cells through the mutation of genes that make the stem cells cancerous; however, that case may not hold for all tumors [21]. The alternative, term tumor-initiating cell, may also cause confusion: a tumor might be initiated by a set of mutations leading to the transformation of one cell type, whereas mutations occurring during the evolution of the tumor might result in another cell type eventually acquiring stem cell properties [5]. Relative rarity is not a defining criterion within the consensus definition of cancer stem cells and is not necessarily a defining feature in all cancer stem cells that have been identified, although cancer stem cells constitute a relatively rare subpopulation of tumor cells in several malignancies [15]. Cancer stem cells are identified and can be isolated based on the expression of specific cell-surface proteins that act as molecular biomarkers, among which the most frequent markers are CD44, CD133, CD24, the epithelial cell adhesion molecule (EpCAM), and the leucine-rich repeat-containing G-protein coupled receptor 5 (LGR5) (Table 1). The expression profiles of those markers are conserved across the cancer cell types of hematogenous and solid tumors.

The cancer stem cell model, thus, contrasts with the stochastic model in that it reflects the hierarchical cellular organization of healthy individuals [22]. These findings established that individual cancer cells are not equal but are instead organized into a cellular hierarchy in which only a rare few leukemia cells can self-renew in a manner reminiscent of the characteristic stem cell properties. This changed the understanding of the underlying biology of cancer, stimulating the exploration of cancer stem cells. However, the phenotypes of cancer stem cells are frequently altered in cancers in a context-dependent manner and are thus suggested to be heterogeneous to be sufficiently flexible to adapt to metabolic reprogramming. Metabolic pathways important for tumor survival are relatively well understood, but the potential for the therapeutic metabolic alteration of cancer stem cells is still being examined. Therefore, the metabolism of cancer cells has garnered researchers’ attention as a critical factor underlying the diversity of cancer stem cells. In particular, increased fatty acid synthesis is frequently observed in different types of cancers, implying that lipogenesis is essential for tumor growth. In this article, we describe the details of cancer stem cell signaling and discuss the therapeutic potential of cancer stem cell-targeted intervention, focusing on fatty acid metabolites and the use of recently emerging single-cell multimodal omics technologies.

## 2. Cancer Cell Signaling and Cancer Stem Cells

Cancer stem cells are identified and can be isolated based on the expression of specific cell-surface proteins that act as molecular biomarkers, among which the most frequent markers are CD44, CD133, CD24, EpCAM, and LGR5 (Table 1). The expression profiles of those markers are conserved across the cancer cell types of hematogenous and solid tumors. Several intracellular signaling pathways are critical for normal stem cell function: the core pathways are the phosphoinositide 3-kinase (PI3K)/protein kinase B (AKT), phosphatase and tensin homolog (PTEN), WNT, and Notch pathways; these promote cell proliferation and, thus, the formation of cancer stem cell-like colonies (Table 2). In cancers, these pathways are frequently altered and are, thus, suggested to be responsible for the fate of cancer stem cells [22,23,24].

The PI3K/AKT pathway functions in a variety of cancers and plays a critical role in the survival and proliferation of healthy cells under physiologic conditions [25,26,27] (Figure 1). For example, prostate cancer development is often associated with the silencing of the tumor suppressor PTEN, a negative regulator of the PI3K/AKT signaling pathway. The PI3K/AKT/PTENPTEN pathway enhances the expression of enzymes involved in de novo fatty acid synthesis, which are indispensable for the cell survival, growth, and metastasis of cancer stem cells. The multifunctional PTEN enzyme inhibits PI3K/AKT signaling in the cytosol, stabilizing the nuclear genome. Various extracellular molecules, including growth factors and/or cell nutrients, act as modulators of the PI3K/AKT/PTEN signaling axis through the activation of receptor tyrosine kinases (RTKs) at the cell membrane toward the NF-κB-mediated gene expression of HIF and MMP9 [28].

WNT-β-catenin pathway is another signaling axis in which fatty acid desaturation could contribute to the maintenance of cancer stem cell function (Figure 1). WNT signaling is transferred into the cytosol via an atypical GPCR, Frizzled, and the low-density lipoprotein receptor-related proteins 5/6 (LRP) and GSK3, with further transduction to the cell nucleus via the β-catenin–T-cell factor and lymphoid enhancer factor (TCF/LEF) complex to enhance the expression of various targets, including Myc, LGR5, and cyclin D1. In contrast, noncanonical WNT signaling is transmitted through Frizzled or other molecules such as receptor tyrosine kinase-like orphan receptors 1/2, phospholipase C, and the Rho family small G-proteins Rac1 and RhoA, from which the signal branches into transcription factors such as NFAT, AP-1, and YAP-TEAD. The canonical WNT pathway acts in a β-catenin-dependent manner and is involved in determining the fate of cancer stem cells; in contrast, the noncanonical WNT pathway is β-catenin-independent and involved in cell polarity formation and cell migration. The Notch receptor family has four members, Notch1 through Notch4, which are processed in a similar manner. The Notch signaling pathway has an important regulatory role in the balance between self-renewal and differentiation. The pathway is activated upon binding of the Notch ligands JAG1, JAG2, and DLL1-4. Binding these ligands triggers the proteolytic processing of Notch by the transmembrane metalloproteinase ADAM-17 [29], with subsequent proteolytic processing by presenilin/γ-secretase, which releases the intracellular domain of the Notch molecule for translocation into the cell nucleus. In the nucleus, the intracellular fragment activates the expression of targets such as the hairy and enhancers of the split family, c-*myc*, NF-κB, cyclin D1, p21, p27, and PPAR, which regulate cell fate, including that of cancer stem cell-like cells [30]. For the effective elimination of cancer stem cells, it is crucial to understand the specific mechanism underlying these complex signaling pathways within the tumor microenvironment. As a representative, the simplified scheme of critical signaling pathways involved in glioma stem cell maintenance was demonstrated in the literature [31].

## 3. Metabolic Reprogramming and the Tumor Microenvironment as a Target of EMT-Induced Cancer Stem Cell Phenotypes

The adaptation of cancer stem cells in a heterogeneous tumor microenvironment, including a low pH, hypoxic conditions, and nutritional state, is a characteristic of cancer stem cells [31,32]. Specific conditions include low pH and acidosis, one of the hallmarks of the tumor microenvironment, which is the consequence of an exacerbated glycolytic metabolism with the reduced removal of acidic waste products, such as lactate [33]. Aberrant metabolic changes occurring in cancer cells generate high concentrations of protons derived from enhanced glucose metabolism. To cope with the accumulation of protons, cancer cells rely on proton exchangers and transporters, which export protons to the microenvironment, allowing malignant cells to survive in the hostile environment that they have created [34]. Therefore, pH regulators are possible therapeutic targets. Carbonic anhydrase, an enzyme responsible for the reversible hydration of carbon dioxide to carbonic acid, is one of the major pH regulators expressed in cancer cells [34]. Inhibitors for carbonic anhydrase, such as sulfonamides and coumarins, display inhibitory effects on breast cancer stem cells [34].

Recently, it has been widely accepted that the adoption of a glycolytic metabolism may be an unforced event that can occur even when oxygen is available in the microenvironment and in cells lacking mitochondrial dysfunction [35]. This reprogramming allows cells to sustain their anabolism while restricting the production of ROS, which are highly toxic for cancer stem cells that are endowed with low levels of ROS detoxification enzymes [35]. In addition to glucose, fatty acids and extracellular catabolites can support cancer stem cell metabolism by providing another fuel pathway [35]. Cancer cells utilize the anabolic process of fatty acid synthesis to derive energy from fatty acid metabolism to support cell growth and proliferation, and the catabolic process of fatty acid oxidation to produce NADH and ATP [36]. In recent years, various drug candidates targeting lipid metabolism, which are likely relevant for cancer stem cells, have entered clinical investigation, including fatty acid synthase inhibitors [35].

The development of secondary tumors at sites in the body distant from the primary tumor is termed metastasis. Although metastasis accounts for the greatest proportion of cancer-associated deaths, it remains the most complex and least understood aspect of cancer biology [37]. In the cancer stem cell model, cancer stem cells must intravasate into the bloodstream or lymphatic system to establish a distant metastasis. To achieve this, cancer stem cells must lose their epithelial characteristics, including epithelial cell junctions and apical–basal cell polarity, and acquire a mesenchymal phenotype, including elongated fibroblast-like morphology and enhanced migration and invasion characteristics. This epithelial mesenchymal transition (EMT) is a physiologic cellular process in which epithelial cells acquire a mesenchymal phenotype and related behavior following the downregulation of their epithelial features triggered by signals the cells receive from their microenvironment [38,39,40,41]. The expression of fatty acid synthase is enhanced through the induction of epithelial mesenchymal transition (EMT) in the metastasis of a certain cancer cell type (Figure 1). The EMT International Association has noted that cancer cells might be able to migrate locally without activating the EMT, possibly using collective migration mechanisms similar to those used during organism development. However, it is unclear whether primary carcinoma cells can complete the entire process of metastatic dissemination without activating, at least transiently, components of the EMT [42]. Despite the continued debate on the involvement of the EMT in cancer progression, the EMT appears to be a major strategy used by carcinoma cells to acquire a cancer stem cell phenotype, making this process an attractive target for novel cancer therapies [43]. However, evidence for the EMT in human cancers has been lacking; thus, an experiment tracking individual cancer cells, from the time they leave a tumor to the point at which they colonize a new organ, is key to concluding the debate [43].

In healthy adult tissues, stem cells reside in niches—microenvironments comprising various cells and ECM components. These niches are discrete and dynamic functional domains that influence stem cell behavior in the maintenance of tissue homeostasis under diverse physiologic and pathologic conditions [44,45]. Stem cell niches have been defined in various tissues, including the intestinal, neural, bone marrow, epidermal, and hematopoietic systems [46,47]. These niches act as sources of signaling molecules that influence the development and self-renewal of stem cells [48]. Cancer stem cells reside in their niches in the same way as normal stem cells but the cancer stem cell niche forms part of the tumor microenvironment, contributing to the genetic and epigenetic heterogeneity of tumor cells [49].

The cancer stem cell model proposes that the elimination of cancer stem cells has great potential as an effective strategy for therapeutically overwhelming malignancies because cancer stem cells residing within tumors are essential to cancer initiation and maintenance, and to driving cancer progression and metastasis. Despite the attractiveness of cancer stem cells as targets, evidence supporting their functionality within the model is limited. However, new findings have shown that cancer stem cells and non-cancer stem cells both have plastic features and are, thus, capable of phenotype transition in response to the tumor microenvironment. Recently, the plasticity of cancer stem cells was shown to hamper the cancer stem cell targeting strategy. Cell ablation experiments in xenografted human cancers [50] with the specific cancer stem cell marker, LGR5, implied that the plasticity of non-cancer stem cells is regulated differently by the tumor microenvironment in the primary and metastatic sites. Thus, cancer stem cells and non-cancer stem cells are thought not to be hardwired but instead, to adapt to certain environmental cues from the tumor microenvironment. These results suggest that the elimination of LGR5-positive cells in metastatic lesions may produce long-lasting therapeutic effects [51]. Metabolic reprogramming to enhance the glucose metabolism is a critical event promoting the EMT-induced cancer stem cell phenotype.

## 4. Therapy Resistance in Cancer Stem Cells

Multiple therapies have been developed to treat cancers; the major treatments are chemotherapy, which uses anticancer drugs to kill cancer cells, and radiation therapy, which uses sufficiently high radiation doses to kill cancer cells and shrink tumors. Other therapies include the surgical removal of cancerous tumors and targeted therapy using molecularly targeted drugs. However, resistance to chemotherapy and radiation is frequently observed, likely because of the intrinsic properties of cancer stem cells acquired through multiple interactions with the tumor microenvironment, including dysregulation of the drug–efflux pump system and DNA repair capacity [52,53,54]. The failure of cancer therapy is a current major concern, with resistance to conventional cancer therapies considered the most serious [55,56]. Resistance can be acquired by various mechanisms. Alteration of the redox balance and the consequent disruption of redox signaling are implicated in the resistance of cancers to chemotherapy and radiation therapy. In particular, the expression of antioxidant enzymes likely contributes to the maintenance of the redox environment, which is advantageous in preventing the development of therapy resistance [55]. The DNA damage caused by ionizing radiation triggers cell cycle arrest at the G1 phase, facilitated by the checkpoint kinases Chk-1 and Chk-2, thus, permitting DNA damage repair. Cancer stem cells in hypoxic regions within tumors are reported to be more resistant to radiation-induced DNA damage than those in normoxic regions, probably because of hypoxia-inducing factors 1α and 2α, which activate the Notch and WNT signaling pathways to maintain the cancer stem cell phenotype.

Chimeric T-cell receptors (CAR) are the cell surface receptors artificially engineered in combination with antigen-binding and T-cell activating functions to target a specific antigen for cell therapy [57]. In CAR T-cell therapy, T cells from patients are cultured ex vivo and engineered such that the modified cells target, attack, and kill cancer cells. CAR T-cell therapy is normally applied to treat certain hematogenous tumors. There are also approaches applying CAR T-cell therapy to solid tumors, including cancer stem cells. The major cell surface cancer stem cell biomarker molecules are CD44 (Pgp-1), CD133 (prominin-1), and the epithelial cell adhesion molecule; naturally, these molecules are promising targets for the development of CAR T-cell therapy. For the treatment of hematogenous tumors, CD33 is the most well-studied target for acute myeloid leukemia (AML)-specific CAR T cells [58]. Preclinical experiments were performed using CAR T-cells in the treatment of solid tumors, including brain and prostate cancers; moreover, clinical trials are also ongoing against CD133 and EGFR (epidermal growth receptor), which are frequently found to be upregulated in these malignant tumors [59].

Immune checkpoint blockade therapy was recently proposed as a promising approach for cancer treatment. This therapy may provide opportunities for the targeted elimination of cancer stem cell subsets to complete the regression of malignant tumors. Cancer stem cells exhibit characteristics that circumvent attacks by the immune system, including altered antigen expression leading to reduced recognition by the immune system. The approach of the immune checkpoint blockade can be used to target cancer stem cells. One major breakthrough in cancer immunotherapy has been the identification of immune checkpoint molecules in tumors, namely cytotoxic T-lymphocyte-associated protein 4 (CTLA-4), the programmed cell death-1 (PD-1), and its ligands programmed death-ligand 1 (PD-L1) and PD-L2 [58]. When T cells were stimulated with PD-1 ligation, a significant elevation of polyunsaturated fatty acids was observed [59]. PD-1 elicits metabolic reprogramming of T cell receptor-stimulated T cells from glycolysis to lipolysis and the utilization of endogenous free fatty acids for β-oxidation. This reprogramming provides a means for the survival of T cells receiving PD-1 signals during their reduced capacity for uptake and utilization of other nutrient classes [59].

## 5. Dietary Intervention Impacting Fatty Acid Metabolites and the Tumor Microenvironment as Potent Regulators of Cancer Stem Cell Signaling

The fatty acid synthesis pathway consists of iterative cycles which elongate two-carbon units in a consecutive manner to form long chain fatty acids (LCFAs). This process begins with the conversion of citrate into acetyl-CoA and oleanolic acid acetate catalyzed by ATP-citrate lyase (ACLY). Acetyl-CoA is then carboxylated to form malonyl-CoA, and then entered into the iterative cycles as the building block for fatty acid synthesis. Monounsaturated fatty acid (MUFA) is synthesized by stearoyl-CoA desaturase (SCD1) and acts as a major component of the plasma membrane. The essential fatty acids, polyunsaturated fatty acids (PUFAs), might be preferentially incorporated by cancer cells from the tumor microenvironment. Conversely, fatty acids are metabolized through the β-oxidation pathway in which ATP is generated as an energy source. The fatty acid is desaturated by an enzyme, fatty acid desaturase, to form a carbon–carbon double bond.

Several reports indicate that cancer stem cells accumulate unsaturated lipids, such as MUFAs, which are precursors for several plasma membrane lipids [60]. The uptake of fatty acids by tumor-infiltrating CD8+ T cells is mediated by CD36 in the tumor microenvironment [61,62], and it enhances the accumulation of free PUFAs in tumor tissues [63]. LCFAs are also accumulated in the tumor microenvironment along with tumor progression [64]. Fatty acid abundance is recognized as a hallmark of a variety of cancers and is attempted to be used as a diagnostic tool and therapeutic in cancer [65] (Figure 1).

Activated fatty acids are not only incorporated into membranes or storage but are also used as substrates to synthesize signaling lipids or for energy production: high activity of this pathway has been reported for aggressive tumor cells and cancer stem cells, especially in nutrient-scarce environments [60]. In addition to its well-known role in energy production, fatty acid metabolism regulates multiple functions of cancer stem cells, including chemoresistance, mainly by suppressing ROS production [61] (Figure 1).

The complexity of foods and eating patterns is well established. Therefore, the assessment of dietary patterns, rather than the traditional reductionist approach focused on specific dietary factors, is a new and more promising strategy when investigating relationships with cancer [66]. Diet is known to impact health, lifespan, disease, and cancer incidence. Altered metabolism is a hallmark of cancer and inspires novel therapeutic strategies; in studies comparing cancer stem cells to non-cancer stem cells, no universal metabolic patterns have emerged [67]. Cancer and non-cancer stem cells preferentially use glycolysis or oxidative phosphorylation depending on the tumor type and model system used, precluding the assessment of microenvironment effects, and the metabolic adaptation of cancer stem cells has emerged as a particularly relevant step during metastatic colonization [67]. Thus, the specific energy requirements of tumor cells, and, notably, of cancer stem cells during metastasis, may represent an opportunity for the treatment of late-stage disease [68]. As nutrients also have pivotal activities in cancer stem cells, clarifying the molecular mechanisms underlying the regulation of the response of cancer stem cells to the diet has recently attracted great interest as a means for cancer prevention and therapy.

A high-fat diet is associated with detrimental effects on the fitness of the stem cell compartment through cell-autonomous or niche-dependent mechanisms [69]. Indeed, high-fat diet-induced obesity augments the number and function of LGR5-positive stem cells [70]. Glioma remains among the deadliest of human malignancies, and the emergence of the cancer stem cell phenotype represents a major challenge to durable treatment response. A study evaluated disease progression in mice fed an obesity-inducing high-fat diet versus a low-fat, control diet, and found that a high-fat diet in hyperaggressive disease accompanied by enrichment of cancer stem cells and shortened survival [71]. Accumulating evidence indicates that cancer stem cells (metastasis-initiating cells) are capable to reactivate oxidative phosphorylation in addition to the glycolytic pathway for surviving in the tumor microenvironment, in which fatty acid is an alternative energy resource to meet the high-fuel consumption in aggressively growing cancer cells. A study demonstrated that experimental metastasis was increased by palmitic acid or a fatty diet, and decreased by the CD36 blockade [72]. Therefore, the coordinative mechanism between fatty acid signaling and the anticancer immune enforcement in cancer stem cells will be of importance for discovering more effective agents targeting cancer stem cells [72].

Calorie restriction is the major nutritional strategy recognized for its effects on stem cell function. The beneficial effects of calorie restriction include enhanced longevity and reduced disease burden. Calorie restriction reduced tumorigenesis in breast tissues and inhibited the self-renewal of stem cells [73] and limited the carcinogenic and metastatic potential of cancer stem cells [73]. Moreover, calorie restriction may be able to indirectly alter the features of cancer stem cells by affecting the gut microbiota, thereby improving immuno-surveillance [74]. Various types of bacteria that are valuable to humans reside in the gut but pathogenic microorganisms are also present [75]. Commensal bacteria may even regulate the effectiveness of immune checkpoint cancer therapy by modulating the immune response to cancers. For example, the benefits of therapy were observed in cancer patients treated with anti-PD-L1, anti-PD-1, and/or anti-CTLA-4 blockade therapies [75]. Microbial imbalance in the gut, called dysbiosis, is now supposed to be one of the gateways to cancer [75]. In addition, the gut microbiota may cause increased efficacy and decreased toxicity of current chemotherapy mediators [76]. The communication between gut microbiota and host cells may indicate a novel research area into potential machinery regulating the efficacy of cancer immunotherapies. In particular, host-directed immunomodulation via the gut microbiota appears promising for a successful defense against cancers. Therefore, the gut microbiota might potentiate not only the effect of an immune checkpoint blockade therapy but also of the so-called onco-checkpoints [76]. Probiotic bacteria may be administered to support cancer treatment. Identifying the relevant supportive mechanisms may provide a promising strategy for probiotic-based dietary therapies, which may be integrated with cancer therapy to improve the outcomes of therapy and patients’ quality of life [76].

## 6. Integrative Healthcare Approach with Emerging Single-Cell Technologies for Precision Nutrition and Functional Medicine

As mentioned above, cancer stem cells exhibit a variety of metabolic reprogramming phenotypes that cannot be completely attributed to the intrinsic heterogeneity of cancer. Given the heterogeneity of cancer stem cells, new strategies with single-cell resolution will become a powerful tool to eradicate the aggressive cell population harboring cancer stem cell phenotypes.

“The fourth paradigm” of data-intensive scientific discovery, presented by the excellent computer scientist Jim Gray in 2009, refers to the intelligent data management techniques and computational systems needed to manipulate and analyze “big data” acquired through research across multiple scientific fields [77]. Under the first paradigm, in the era of Aristotle, only experimental science with empirical methods existed. Then, theoretical science occurred through the effort of the scientists such as Kepler, Newton, and Maxwell in the second paradigm. The third paradigm emerged in the mid-20th century through the work of John von Neumann’s large-scale computational simulation. The fourth paradigm provides an integrating framework that allows the first three paradigms to interact and reinforce each other. In the field of life sciences, the fourth paradigm enables us to fully comprehend the biological processes performed by complex networks of biomolecules’ interactions and gene regulation programs at the single-cell level. Therefore, we can present single-cell technologies for transcriptomic, epigenetic, and metabolomic profiling that can contribute to distinct facets of cancer research. We focus mainly on studies concerning human single-cell transcriptomics and epigenomics and address newly developed technologies that bridge transcriptomics and metabolomics at the single-cell level. Finally, we highlight attempts to apply these single-cell multimodal omics methodologies to deeply personalized phenotyping and personalized diagnosis for precision food nutrition and functional medicine [78]. Bioinformatics emerged from the human genome project and enabled large-scale DNA sequencing data using next-generation sequencing technologies. These technologies allowed the systematic understanding of the gene expression profiles that were not only restricted to protein-coding genes, but also include non-coding RNAs (ncRNAs) such as microRNA (miRNA) and long non-coding RNAs (lncRNAs), the regulatory elements in 5’- and 3’- untranslated regions (UTRs), and post-transcriptional RNA modification events (RNA editing) including nucleobase modifications, such as cytidine to uridine and adenosine to inosine deaminations as well as nucleotide additions and insertions [79,80,81,82,83,84]. For example, miR-375 deregulation disturbs several critical cellular pathways, particularly cell cycle regulation in prostate cancer cells [85]. These herald the shift to next-generation systems biology, which aims to reconstruct the interactions to explain the synergistic network in the human body by utilizing machine learning [86].

Various technological inventions are associated with single-cell technologies. First are the (1) approaches to examine genome-wide chromatin accessibility of thousands of cells at the single-cell resolution (e.g., scATAC-seq method) [87]. Second, (2) there is a multimodal single-cell phenotyping method developed in the Technology Innovation laboratory at the New York Genome Center, called Cellular Indexing of Transcriptomes and Epitopes by Sequencing (CITE-seq) [88]. This method integrates cellular protein measurements and transcriptome analysis by high-throughput single-cell RNA sequencing (scRNA-seq) into an efficient, single-cell readout. (3) RNA expression and protein sequencing assay (REAP-seq) [89] is also a recently developed multimodal single-cell methodology, which enables deeper characterization of cellular subtypes and functional states by measuring the expression of cellular proteins and RNA. The bidirectional relationship between dietary nutrition and the immune system is pivotal for maintaining health. Determining the full range of cells within the human body has traditionally been difficult, owing to technical limitations restricting the number of parameters that can be measured simultaneously and the minimum amount of starting material required [90]. Single-cell approaches overcome this restriction through their unbiased nature of assessment; indeed, they are shedding light on previously unappreciated heterogeneity within immunity [90]. Combined with these analytical approaches, computational methods for the analysis and integration of single-cell omics data across different modalities will open new avenues for the accurate reconstruction of gene-regulatory and signaling networks driving cellular identity and function [91,92,93]. We found that certain types of diets are good therapy for chronic diseases and have beneficial effects on the risk factors [94,95]. The human body consists of 37 trillion cells, and, in 2016, an international collaborative project, The Human Cell Atlas, was established to map all the cells in the human body and their relationships with each other [96]. A comprehensive lipidome atlas by tandem mass spectrometry information was also reported [97]. Therefore, along with these large global projects, integrative healthcare approaches with emerging single-cell technologies will provide in-depth insights into current research on precision nutrition and functional medicine and allow the discovery and development of novel therapeutic approaches and methods for cancer prevention in the near future.

## 7. Conclusions and Perspective

This article provides an overview of the most important signaling pathways of cancer stem cells with their relevance to the EMT and tumor microenvironment. We also suggest emerging strategies within cancer immunotherapies targeting cancer stem cells as protection from tumor recurrence. Diversity within tumor cells at the genetic and functional levels and their coexistence with the microenvironment also increases tumor fitness, enabling tumor cells to offset the survival pressures imposed by therapy [98]. Fatty acid metabolites that target cancer stem cells should also be a topic for future study [99]. In addition, metabolomics of the tumor microenvironment surrounding cancer stem cells may contribute to the elucidation of the stemness of cancer stem cells. Therefore, these aspects of the regulatory mechanism should be reflected in cancer therapy [100]. By targeting metabolically diverse populations and considering the tumor microenvironment, more effective therapies and novel ways of targeting the lipid metabolites in the tumor microenvironment of cancer stem cells may be promising in the future.

## Figures and Tables

**Figure 1 metabolites-13-00709-f001:**
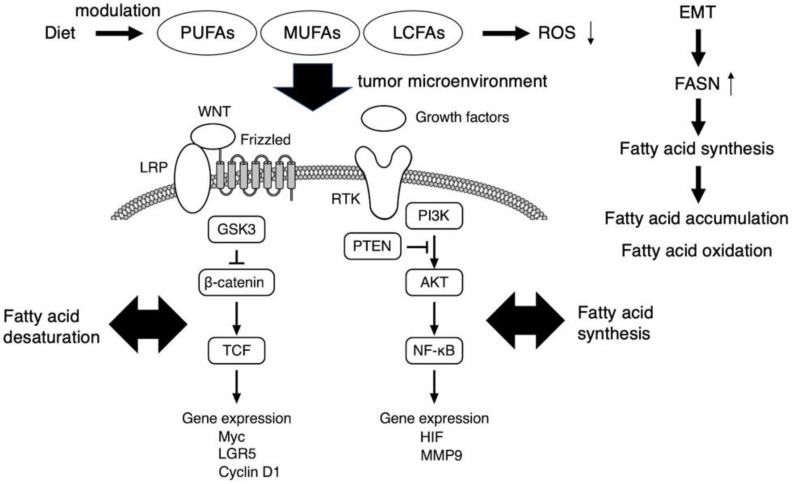
The role of fatty acids as potent regulators of cancer stem cell signaling. PI3K/AKT and WNT-β-catenin axis, which associate with fatty acid synthesis and desaturation, respectively, are critical in cancer stem cell signaling. The FASN expression is enhanced upon EMT, and thereby cancer stem cells accumulate MUFAs and LCFAs which are used for aberrant cell proliferation and also act as signaling metabolites. Free PUFA accumulation is observed in tumor tissues. Fatty acid metabolism regulates multiple functions of cancer stem cells, including chemoresistance, mainly by suppressing ROS production. Therefore, diet-based therapies targeting the lipid metabolites in the tumor microenvironment of cancer stem cells may be promising. EMT, epithelial to mesenchymal transition; FASN, fatty acid synthase; LCFA, long chain fatty acid; MUFA, monounsaturated fatty acid; PUFA, polyunsaturated fatty acid; ROS, reactive oxygen species.

**Table 1 metabolites-13-00709-t001:** Markers for cancer stem cells.

Name	Other Name	Type	Function in Normal Cells
CD44	Pgp1	type I transmembrane	receptor for hyaluronan [11]
CD133	prominin-1	pentaspan transmembrane	costimulatory factor of T cells [11]
CD24	heat stable antigen	GPI-anchored	costimulatory factor of T cells [12]
CD326	EpCAM	GPI-anchored	cell adhesion [16]
LGR5	GPR49	GPCR	receptor for R-spondin [4]

**Table 2 metabolites-13-00709-t002:** Signaling molecules of cancer stem cells.

Name	Type	Function in Normal Cells
PI3K	protein kinase	cell survival and proliferation
PTEN	protein phosphatase	tumor suppressor
WNT	secreted glycoprotein	Frizzled binding
Notch	type I transmembrane	development and differentiation

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
