# Peer review of "Fatty Acid Metabolites and the Tumor Microenvironment as Potent Regulators of Cancer Stem Cell Signaling"

_metabolites, 2023, doi:10.3390/metabo13060709_

Round 1

Reviewer 1 Report

This manuscript provides an overview of the cellular hierarchy of cancer cells, highlighting the importance of rare leukemia cells with self-renewal properties. The PI3K/AKT pathway is critical for the survival and proliferation of healthy cells, but also plays a role in various cancers. Additionally, cancer stem cells exhibit metabolic reprogramming phenotypes that cannot be fully attributed to intrinsic heterogeneity. With the heterogeneity of cancer stem cells, new strategies with single-cell resolution can be a powerful tool to eradicate aggressive cancer stem cell phenotypes. The article suggests valuable strategies in cancer immunotherapy to inhibit tumor recurrence by targeting signaling pathways of cancer stem cells within the tumor microenvironment. The manuscript is well-organized and clearly written. Once the necessary corrections have been made, it should be accepted for publication.

1.     It would be better to list the references at Table 1.

2.     Including graphs to summarize the signaling molecules of cancer stem cells can enhance the clarity and impact of the article, making it easier for readers to understand the complex signaling pathways involved.

Author Response

Reviewer 1

 It would be better to list the references at Table 1.

Following the suggestion, the references were added to Table 1.

 Including graphs to summarize the signaling molecules of cancer stem cells can enhance the clarity and impact of the article, making it easier for readers to understand the complex signaling pathways involved.

As is commonly known, cancer stem cells (CSCs) elicit complexed signaling pathways within the specific tumor microenvironment. Therefore, for effective elimination of CSCs, it is crucial to understand the specific mechanism to each type of CSCs. Thus, it is beyond the scope of this review article to describe an overview of CSC signaling by showing a generalized scheme including graphs. Instead, a very important paper in which the authors summarized the critical signaling pathways involved in glioma stem cell maintenance has been cited (reference no. 33 of the revised manuscript). We thank the reviewer for carefully reading our manuscript and providing us constructive comments to improve the manuscript.

Reviewer 2 Report

Yoshiyuki Murai and Satoru Matsuda's review is an overview of cancer stem cells. 

The study is interesting, and the issue is important to understanding tumor pathogenesis and the development of therapeutic strategies.

However, in this form, the manuscript needs to be significantly improved.

1- the title  "Fatty acid metabolites and the tumor microenvironment as potent regulators of cancer stem cell signaling" only partially reflects the review's overall content. In particular, the role of fatty acids as regulators of cancer stem cells must be significantly improved.

2- the organization of the work must also be revised. Some concepts in paragraph 3 must be moved to paragraph 1, as they are general concepts of stem cells.

3-paragraph 2 must be improved. The molecular signalings are just listed in the text. I suggest improving the text and table 1 with more columns to describe the role of those molecules in cancer stem cells. Alternatively, the authors may add a representative scheme. 

Author Response

Reviewer 2 

the title  "Fatty acid metabolites and the tumor microenvironment as potent regulators of cancer stem cell signaling" only partially reflects the review's overall content. In particular, the role of fatty acids as regulators of cancer stem cells must be significantly improved.

The updated information regarding the role of fatty acids as regulators of cancer stem cells has been added (page 10, line 22 to page 11, line 3). An important Nature article concerning this issue has been cited (reference no. 74 of the revised manuscript).

the organization of the work must also be revised. Some concepts in paragraph 3 must be moved to paragraph 1, as they are general concepts of stem cells.

Following the advice, some concepts in paragraph 3 of the original manuscript describing the general concepts of cancer stem cell has been moved to paragraph 1 (page 2, line 25 to page 3, line 2 of the revised manuscript).

paragraph 2 must be improved. The molecular signalings are just listed in the text. I suggest improving the text and table 1 with more columns to describe the role of those molecules in cancer stem cells. Alternatively, the authors may add a representative scheme. 

Cancer stem cells (CSCs) elicit complexed signaling pathways within the specific tumor microenvironment. Therefore, for effective elimination of CSCs, it is crucial to understand the specific mechanism to each type of CSCs. Thus, it is we think beyond the scope of this review article to describe more the role of those molecules in each cancer stem cell type. Instead, a critical paper that summarizes the critical signaling pathways involved in glioma stem cell maintenance has been cited (reference no. 33 of the revised manuscript). We thank the reviewer for allowing us to significantly improve the manuscript.